# Deep Learning for Cyberbullying Detection: A GloVe-based Comparative Analysis of CNN and LSTM Models

## Abstract

Social networks were created to fulfill human needs, driven by people's eagerness to learn new things and stay informed about global events. To detect cyberbullying on social media, this research compares two deep learning architectures: GloVe+CNN and GloVe+LSTM. Textual data were represented using pre-trained GloVe embeddings, and CNN and LSTM were employed as classification layers to identify sequential and local patterns, respectively. Using a multiclass cyberbullying dataset, the models were trained and evaluated. The results show that although both architectures perform well, GloVe+LSTM outperforms CNN in terms of F1-score and recall, indicating better contextual understanding. The experimental results demonstrate the superiority of LSTM, in terms of accuracy.

## 1 Introduction

Cyberbullying has become a major issue due to the rapid rise of social media platforms, mainly because of the psychological and emotional harm it inflicts on victims. In February 2025, 99.6% of Saudi Arabians used at least one social media site, and 99% of the country's population had Internet access Abdurakhmonova (2025), Alhassan et al. (2025).

It is now essential to automatically identify harmful content to protect users and keep online environments safe. Since abusive language is subjective, lacks clear signals, and often involves loud or intentionally hidden content (such as disguised slurs), detecting cyberbullying remains difficult. Lu et al. (2020). Traditional machine learning (ML) techniques have addressed this problem extensively, but these methods often require manual feature engineering, which is time-consuming and error-prone. Additionally, they have trouble understanding the full context of violent language and struggle to generalize across different platforms.

Paul et al. (2022). On the other hand, deep learning (DL) models like CNNs, LSTMs, and GRUs, as studied by Cho et al. (2014) have demonstrated superior contextual awareness and automatic feature extraction skills, especially when combined with semantic word embeddings like GloVe. Buan & Ramachandra (2020). ). Goals and research gap: There is also a lack of systematic comparisons between different DL architectures applied to the same dataset, despite previous research showing the superiority of DL-based techniques over traditional ML models in cyberbullying detection. Specifically, there is a lack of research comparing CNN-based and LSTM-based models in a single experimental setup to evaluate their performance in multiclass classification tasks. To address this gap, this study uses a common benchmark dataset for cyberbullying detection to assess and compare two deep learning architectures: GloVe+CNN and GloVe+LSTM.

Goals and Research Gap: There is a lack of systematic comparisons between different DL architectures applied to the same dataset, despite previous studies showing that DL techniques outperform traditional ML models in cyberbullying detection Cho et al. (2014), Gers et al. (2000), Huang et al. (2015), Specifically, research comparing CNN-based and LSTM-based models Caroppo et al. (2020), Banerjee et al. (2019) within a single experimental setup to evaluate their performance in multiclass classification remains limited. To fill this gap, this research uses a common benchmark dataset for cyberbullying detection to evaluate and compare two deep learning architectures: GloVe+CNN and GloVe+LSTM.

However, few studies have directly compared fundamental deep learning architectures under consistent experimental conditions. Unlike previous work that focused on hybrid or task-specific architectures, our study provides a systematic benchmark comparison between two core deep learning models: GloVe+CNN and GloVe+LSTM. This analysis offers insights into their respective strengths in modeling localized textual features versus long-range dependencies. Few studies, however, have explicitly contrasted basic deep learning architectures in reliable experimental setups. Unlike earlier research emphasizing hybrid or task-specific designs, our study presents a thorough benchmark comparison of GloVe+CNN Pennington et al. (2014) and GloVe+LSTM, Rafiq et al. (2018). two fundamental deep learning models. This analysis highlights their relative benefits in modeling long-range dependencies versus localized textual properties. To compare GloVe+CNN and GloVe+LSTM models on multiclass cyberbullying detection, we develop a comparative assessment framework.

We compare the effectiveness of sequential and convolutional modeling by evaluating their performance using standard metrics such as accuracy, precision, recall, and F1-score. We include confusion matrix-based error analysis to identify specific flaws and limitations of each model, fostering a deeper understanding of misclassification patterns. This paper is organized as follows: Section 2 reviews related deep learning models used for cyberbullying detection. Section 3 describes the dataset, preprocessing steps, and evaluation methodology. Section 4 details the proposed models and presents experimental results and summarizes the key findings and suggests directions for future research. In the next section, we examine existing approaches to cyberbullying detection, highlighting their strengths and weaknesses.

## 2 BACKGROUND AND RELATED WORK

A growing threat on social media, cyberbullying has significant emotional impacts on its victims. Due to the implicit, contextual, or complex language used, it is often hard to recognize and appears as hostile, degrading, or threatening content. Education about online behavior, emotional self-management training, and awareness campaigns are some prevention and intervention methods introduced to address this issue Cheng et al. (2019),Brownlee (2016) . However, these strategies often fail, especially during adolescence when peer pressure is strongest Ho et al. (2019), and studies show that traditional school-based programs usually lead to only modest behavioral changes Dadvar & Eckert (2020) , Kowsari et al. (2019)). Researchers are increasingly turning to automated detection technologies to support prevention efforts. Early approaches relied mainly on hand-made features and rule-based or traditional machine learning methods. For example, Iwendi et al. (2023) used various machine learning models to classify hate speech on large datasets from platforms like YouTube, Twitter, and Reddit, with mixed results. Additionally, Van Hee et al. created a Dutch social media corpus with annotations for different bullying roles—victim, aggressor, and by stander enabling multi-label classification using machine learning techniques. Deep learning (DL) models have recently shown improved performance in cyberbullying detection. Using the DISCo Kaggle dataset, Iwendi et al. (2023). and Murshed et al. (2022) evaluated Bi-LSTM, GRU, and RNN models; Bi-LSTM achieved the highest accuracy at 82.18%. To improve detection results, many researchers have proposed hybrid models that combine different DL architectures. For example, Alotaibi et al. (2021) and Raj et al. (2021) achieved 88% accuracy on Twitter comments with a multichannel model that integrated CNN, Bi-GRU, and transformers. DEA-RNN, a model combining optimization techniques with recurrent networks, was introduced by Murshed et al. (2022), Mehendale et al. (2022) compiled a dataset of 11,000 tweets and assessed the effectiveness of various machine learning methods, including SVM, LTSM, CNN + LTSM, GRU, and CNN + GRU. The results show that all four DL models outperform SVM in detecting hostile tweets. CNN-LSTM identified religious hate speech with the highest F score of 59%. Duwairi et al. (2021) evaluated the performance of CNN, CNN-LSTM, and BiLSTM-CNN models for automatically classifying hateful content. The highest prediction accuracy, 74%, was achieved by CNN and BiLSTM-CNN. They also proposed an SVM-based method for cyberbullying detection and compared its prediction accuracy with the NB classifier. The results showed that SVM, with a score of 95.74%, outperformed NB in identifying cyberbullying.

Despite recent advances, few benchmarks compare different DL architectures using the same datasets and evaluation criteria. It is hard to evaluate the strengths of core architectures like CNN and LSTM individually, as most publications focus on innovative model combinations or their application to various datasets. Our work offers a systematic comparison of two popular deep learning

architectures, GloVe+CNN and GloVe+LSTM, under identical benchmark conditions, unlike previous studies that mainly assess hybrid or task-specific models. This allows us to quantitatively analyze the tradeoffs between sequential modeling (LSTM) and local pattern recognition (CNN) in multiclass cyberbullying detection. By factorizing a global co-occurrence matrix, the unsupervised algorithm GloVe (Global Vectors for Word Representation) produces word embeddings that capture semantic patterns across a corpus. GloVe's portability and availability as pretrained models make it highly useful; for example, it remains supported by libraries such as spaCy and is used in applications ranging from identifying psychological discomfort to measuring semantic similarity. GloVe continues to serve as a reliable baseline for tasks like sentiment analysis and fake-news classification, providing efficiency and interpretability despite the growth of contextual models. Hauschild & Eskridge (2024) demonstrate GloVe's ongoing relevance in modern designs through recent research incorporating it into hybrid frameworks, such as using graph neural networks or self-supervised techniques. Table 1 summarizes recent research efforts on cyberbullying detection across different social media platforms. The studies vary in methodology, including traditional prevention techniques, machine learning models, and advanced deep learning architectures. While some focus on annotated datasets and behavioral analysis, others propose hybrid models to improve accuracy and training efficiency. Notably, deep learning models like Bi-LSTM and transformer-based architectures have demonstrated superior performance in accuracy, precision, recall, and F1-score. Building on these insights, we present our proposed methodology in the next section.

Table 1: Summary of recent studies on cyberbullying detection techniques.

| Authors | Contribution | Methods / Models | Platform / Dataset | Performance |
|---|---|---|---|---|
| Dadvar et al. (2012) | Hate speech detection across platforms | ML with feature analysis | YouTube, Reddit, Wikipedia, Twitter | 39% in precision, 6% in recall, 15% F-measure |
| Kowsari et al. (2019) | Annotated Dutch corpus with bullying roles | ML classifiers with threat/insult annotation | Dutch social media posts | Accuracy: 88.45%, Precision: 89.35%, Recall: 89.74% |
| Alotaibi et al. (2021) | Multichannel DL model for Twitter | CNN + Bi-GRU | Twitter | Accuracy: 88% |
| Murshed et al. (2022) | Hybrid DEA-RNN approach for fast training | DEA + Elman RNN vs. Bi-LSTM, RNN | Twitter | Accuracy: 90.45%, Precision: 89.52%, Recall: 88.98%, F1-score: 89.25% |
| Iwendi et al. (2023) | Comparison of DL models | Bi-LSTM, GRU, LSTM, RNN | DISCo (Kaggle) | Bi-LSTM: 82.18% Accuracy |
| Hadi et al. (2024) | Evaluated ML models using the Bag of Words and feature selection across six categories | SVM, Logistic Regression, Naïve Bayes, KNN, and Random Forest. | Not specified | SVM and Logistic Regression 83% accuracy, Naïve Bayes 62% accuracy, KNN and Random Forest 75% and 81% Accuracy respectively, |
| Sharif et al. (2024) | Perform Sentence Level Sentiment Analysis (SLSA) ML techniques | TF-IDF and machine learning algorithms | Twitter | 90% Accuracy, 82% Precision, 74% Recall, and 78% F1-score |
| SAY (2025) | combining (ML) classifiers with (NLP) techniques | Random Forest, Support Vector Machine, Logistic Regression, Naïve Bayes, and K-Nearest Neighbor. | Twitter | Accuracy rates of 94 %, 93 %, 92 %, 92 %, and 73% respectively |
| Mahdi et al. (2025) | hybrid deep learning model called Robustly Optimized Bidirectional Encoder Representations from the Transformers with the Bidirectional LSTM-based Attention model (RoBERTa-BiLSTM) | BERT-base, RoBERTa-base, RoBERTa-large, DistilBERT, ALBERT-xxlarge, XLNet-large, ELECTRA-base, DeBERTa-v3 | Not specified | Accuracy of 94.8%, Precision of 96.4%, Recall of 95.3%, F1-score of 95.8%, and an AUC of 98.5%. |

# 3 PROPOSED METHODS

## 3.1 DATA SET

TThe Cyberbullying Classification dataset, which includes tweets categorized into five groups: religion, age, gender, ethnicity, and non-cyberbullying, was used in this study. Before analysis, the tweets were cleaned and preprocessed by removing stop words, replacing slang definitions, and

lemmatizing words to their basic forms. Three dataset splits were created: 90% for training, 5% for validation, and 5% for testing. An initial exploratory analysis was conducted to assess class distribution. Figure 1 shows a balanced dataset with a significant portion labeled as " non-cyberbullying." To ensure reliable classification results, this imbalance was considered during the modeling process. We now describe the experimental setup and evaluation criteria used to validate our models.

## 3.2 FEATURE EXTRACTION

For feature extraction, the dataset is organized into five labeled categories representing different types of cyberbullying, along with a non-cyberbullying class. The labels are defined as follows: 0 = Religion, 1 = Age, 2 = Gender, 3 = Ethnicity, and 4 = Not Cyberbullying. Figure 1 shows the sentiment distribution across these categories, which remains fairly balanced. Specifically, Gender and Ethnicity each make up 23.0% of the data, while Religion and Age account for 20.3% and 20.0%, respectively, and the Not Cyberbullying class accounts for 19.0%. This distribution ensures that the feature extraction process benefits from a representative and diverse set of samples, reducing class imbalance and improving the robustness of the extracted features for downstream classification tasks. A feature extraction method was used in this study's context of cyberbullying detection, and a brief overview of this method is provided.

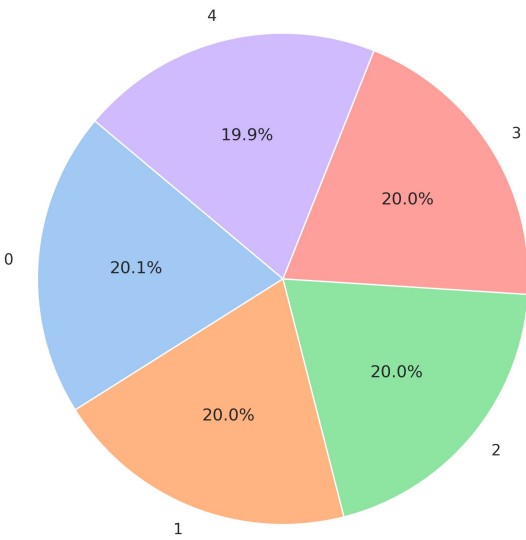

Figure 1: Sentiment distribution

### GLOBAL VECTORS FOR WORD REPRESENTATION

The concept of topic modeling has been explored through various statistical models. Latent semantic analysis is a well-known method in this area that extracts meaningful semantics using matrix factorization Karim et al. (2022). While effective, it has some limitations compared to Word2Vec. GloVe was later introduced to create a more effective model that combines the strengths of both approaches. GloVe often outperforms Word2Vec. It uses a word context matrix and a word co-occurrence matrix to analyze the entire corpus.

## 3.3 PRE-TRAINED EMBEDDINGS: GLOVE EXPLANATION

Before exploring model architectures, we added pre-trained word embeddings to improve our text data representation. Specifically, we used GloVe (Global Vectors for Word Representation), a popular technique developed by Stanford that captures semantic and syntactic relationships between words based on co-occurrence statistics from large texts. Each word is mapped into a 300-

dimensional vector space where similar words are closer together. In our setup, GloVe embeddings initialized the embedding layer of both CNN and LSTM models. These embeddings remained non-trainable to preserve their pre-learned semantic structure. This approach allows the models to leverage general language understanding without additional training on word meanings, which is particularly useful when working with small datasets.

## 3.4 METHODOLOGY

For multi-class cyberbullying detection using social media text data, we used two deep learning models in this study: a Convolutional Neural Network (CNN) and an LSTM network. The dataset includes labeled text samples from five sentiment categories: religion, age, gender, ethnicity, and non-bullying.

Lowercasing, tokenization, and padding to a fixed sequence length of 300 were among the preprocessing steps performed on the data. Pre-trained 300-dimensional GloVe vectors, which provide semantically rich word representations, were used to initialize the word embeddings. To preserve their semantic integrity, these embeddings remained non-trainable and were used in the embedding layers of both CNN and LSTM models. A 1D convolutional layer, batch normalization, max pooling, and two dense layers with dropout for regularization were incorporated into the CNN model. In contrast, the LSTM model contained a bidirectional LSTM layer followed by dense layers. Both models concluded with a softmax activation layer for multiclass classification. The Adam optimizer and categorical cross-entropy loss function were used to train the models over 50 epochs. Evaluation measures included accuracy, precision, recall, and F1-score, with training, validation, and test splits maintained.

# 4 RESULTS AND DISCUSSION

We evaluated the performance of the GloVe+CNN and GloVe+LSTM models on a multiclass cyberbullying detection dataset that includes five categories: religion, age, gender, ethnicity, and not bullying. The classification results were measured using accuracy, precision, recall, and F1-score. Both models outperformed a baseline logistic regression model with TF-IDF features, confirming the benefit of deep learning in capturing complex language patterns. Among the two, GloVe+LSTM achieved the highest overall accuracy and F1-score, particularly in identifying minority and semantically ambiguous classes. The simple (unweighted) average is calculated after the macro average evaluates the metric separately for each class. Since it assigns equal weight to each class regardless of size this demonstrates that LSTM consistently outperforms CNN across all classes. The weighted average is then computed by considering the number of samples in each class. This shows that, even when accounting for class sizes, LSTM outperforms CNN overall. Because macro and weighted scores often differ significantly in imbalanced datasets, values close to the macro average suggest the dataset is well balanced. Figures 2 and 3 display the confusion matrices for the CNN and LSTM models, respectively.

- The CNN model performed well on the non-bullying class, correctly identifying 525 instances, but struggled with gender and ethnicity, often misclassifying them as age or religion. For example, 11 instances of ethnicity were mistaken for gender, highlighting challenges in distinguishing between similar discriminatory contexts.
- The LSTM model, in contrast, exhibited better generalization across minority classes. It accurately classified 409 non-bullying instances and showed improved differentiation between age and ethnicity (only three misclassified between them), thanks to its ability to model long-range dependencies in text.

This confirms that LSTM is more effective at capturing contextual information, while CNN tends to depend on local n-gram patterns, making it more susceptible to overlap between semantically similar labels. Although the LSTM model achieved higher accuracy and robustness across categories, it required longer training times and more computational resources. The CNN model, though slightly less accurate, trained faster and performed well on dominant classes — making it more suitable for real-time or resource-limited applications. Additionally, the GloVe embeddings played a key role in both models' success by capturing semantic similarities between words, thereby improving feature representation

Table 2: Comparaison des performances des modèles LSTM et CNN pour la détection de cyber-harcèlement.

| Classes | Model | Precision | Rappel | F1-score | Support |
|---|---|---|---|---|---|
| Religion | LSTM | 0.94 | 0.96 | 0.95 | 391 |
| | CNN | 0.94 | 0.95 | 0.94 | 391 |
| Age | LSTM | 0.96 | 0.99 | 0.97 | 407 |
| | CNN | 0.97 | 0.97 | 0.97 | 407 |
| Gender | LSTM | 0.87 | 0.81 | 0.84 | 373 |
| | CNN | 0.63 | 0.92 | 0.75 | 373 |
| Ethnicity | LSTM | 0.96 | 0.97 | 0.97 | 421 |
| | CNN | 0.96 | 0.97 | 0.97 | 421 |
| Not Cyberbullying | LSTM | 0.80 | 0.81 | 0.80 | 402 |
| | CNN | 0.86 | 0.48 | 0.61 | 402 |
| **Accuracy** | CNN | | | 0.91 | |
| | LSTM | | | 0.94 | |
| **Macro Avg** | LSTM | 0.91 | 0.91 | 0.91 | 1994 |
| | CNN | 0.87 | 0.86 | 0.85 | 1994 |
| **Weighted Avg** | LSTM | 0.91 | 0.91 | 0.91 | 1994 |
| | CNN | 0.88 | 0.86 | 0.85 | 1994 |

For the religion class, the models with excellent performances, with an F1 of 0.95 for LSTM and 0.94 for CNN. Please note that the voice association in this category is fully captivated by the representation. The age class is the most suitable for these models, with an F1-score of 0.97 . The confusion is almost instantaneous, which confirms that the text indicators appear on the sails and are distinct.

The gender class is more difficult. The CNN get a higher altitude (0.92) but a reliable precision (0.63), which indicates that it tends to the class of gender, generating a number of faux positives. The LSTM, in repair, is more conservative, with a precision of 0.87, but a maximum speed limit of 0.81, which will increase the size of the price. The confusion matrices monitor confusion and differences without bullying and ethnicity. For the ethnicity class, the models perform well, with an F1 of 0.97 for LSTM and 0.94 for CNN. The long-term memory of the LSTM architecture completely captures the linguistic nuances of this category. In fact, the class not bulling F1 de 0.80 for LSTM and 0.87 for CNN, but this remains a source of important confusion, especially for the gender class. This can be expanded by the legal proximation of certain non-offensive expressions with a sensible content. These results suggest that CNN is a performant on the classes with lexical structures, thanks to the capacitance of capturing local motifs. The LSTM, from its heart, is more complex in complex classes such as gender and ethnicity, and its differences in a long period of time are important. A hybrid application combining CNN and LSTM will allow you to benefit from advanced architectures.

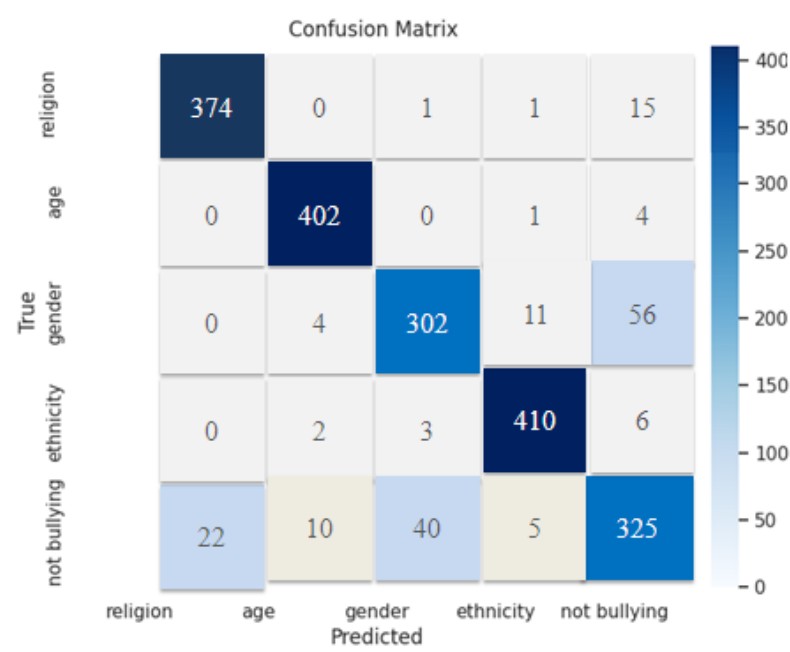

Figure 2: Confusion Matrix for CNN Model

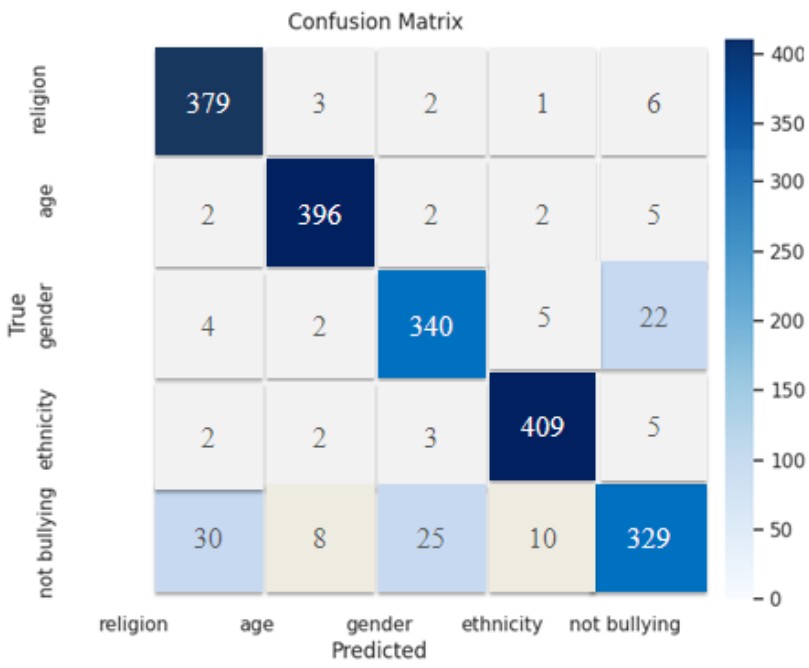

Figure 3: Confusion Matrix for LSTM Model

## 4.1 Limitations and Observations

Several limitations were observed:

1. Class imbalance affected performance, particularly in the ethnicity and gender categories, where misclassifications were most common.

2. Both models had difficulty detecting subtle or implicit bullying language, especially when users employed slang, sarcasm, or obscured spelling.

3. Our experiments were conducted on a single dataset; testing on other platforms (e.g., Reddit, TikTok) remains to be done.

## 4.2 conclusion and future works

The combination of quantitative metrics, confusion matrices, and error analysis provided a deeper understanding of each model's strengths and weaknesses. While LSTM demonstrated better overall performance, CNN remains a strong alternative in limited environments. These insights can guide future improvements, including data augmentation, class balancing strategies, or transformer-based models. Finally, we summarize our findings and suggest directions for future research. In this study, we performed a systematic comparison between two deep learning models GloVe+CNN and GloVe+LSTM for multiclass cyberbullying detection. Unlike previous studies that focus on hybrid architectures, our approach evaluates these two core architectures under consistent experimental conditions, providing valuable insights into their respective strengths. Our results show that GloVe+LSTM outperforms GloVe+CNN in both accuracy and generalization, especially for minority classes such as ethnicity and gender. However, the CNN model demonstrated faster training times and competitive performance, making it suitable for real-time applications. This work improves understanding of how different architectures handle nuanced and imbalanced abusive content. It also highlights the importance of sequence modeling in detecting implicit online aggression. For future work, several avenues for improvement can be considered. First, the integration of more recent Transformer-based models, such as BERT or RoBERTa, could better capture the contextual subtleties of language and improve the detection of cyberbullying. Second, the exploration of hybrid approaches combining CNN and LSTM would offer the possibility of taking advantage of both the capture of local features and sequential dependencies. Furthermore, the use of class rebalancing techniques or synthetic data generation would be relevant to reduce the impact of imbalances observed between certain categories. Finally, expanding the corpus to multimodal data (text, image, video) would represent an important step towards more robust detection systems adapted to real social media environments.

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
