# OpenReview forum: "Deep Learning for Cyberbullying Detection: A GloVe-based Comparative Analysis of CNN and LSTM Models"
_ICLR.cc/2026/Conference — Submitted to ICLR 2026_

### Official Review · Reviewer_umYY · 2025-10-28

**Soundness:** 1
**Presentation:** 1
**Contribution:** 1
**Rating:** 2
**Confidence:** 5

**Summary:**

An unfinished and not ready for submission work

**Strengths:**

The topic is worth to investigate

**Weaknesses:**

Lack of novelty.: The combination of GloVe embeddings with CNN or LSTM for cyberbullying detection has been extensively explored in 2017–2021 literature. The proposed work offers no algorithmic, architectural, or theoretical innovation beyond re-running baselines.

Outdated modeling choice: In 2025, contextual transformers (BERT, RoBERTa, DeBERTa, etc.) are the accepted baselines. Evaluating only static GloVe embeddings limits the paper’s relevance to the current ICLR audience.

Contribution limited to evaluation: The main contribution lies in re-implementation and performance comparison rather than any new methodological insight. There is no proposed loss function, data augmentation, interpretability method, or cross-platform generalization.

Presentation and editorial problems: Several tables and figures appear unfinished or poorly formatted (e.g., Table 2 title written in French; figures without proper captions or axis labels). The narrative contains repetitive sections (duplicated “Goals and Research Gap”), grammatical inconsistencies, and typographical errors.

Insufficient discussion of ethical and societal context: Cyberbullying detection involves sensitive language and fairness aspects, yet the paper lacks any discussion of bias, explainability, or data ethics.

Limited comparison baseline: The only baseline is logistic regression + TF-IDF; no comparison with transformer-based or hybrid architectures is provided, making the claimed performance improvements trivial.

**Questions:**

Please check the weakness section

---

### Official Review · Reviewer_pKdd · 2025-10-30

**Soundness:** 2
**Presentation:** 2
**Contribution:** 1
**Rating:** 2
**Confidence:** 4

**Summary:**

The paper compares two classic deep-learning setups for spotting different kinds of cyberbullying in tweets: a CNN and a bidirectional LSTM, both using fixed 300-d GloVe embeddings. With the same preprocessing and training recipe, the BiLSTM comes out a bit more accurate overall and makes fewer mix-ups between sensitive categories (like gender vs. ethnicity), while the CNN runs faster and does fine on more obvious cases. The analysis leans on per-class F1s and confusion matrices to show where each model shines or stumbles. The paper discusses limitations including single dataset, difficulty with subtle/implicit bullying and suggests future work like transformers, hybrid CNN–LSTM, class rebalancing, multimodal data.

**Strengths:**

1.	 The paper gives a focused, head-to-head comparison of two canonical DL architectures under a single, consistent setup for multiclass cyberbullying—useful because much prior work either mixes datasets or evaluates hybrids without isolating core trade-offs.

2.	 The paper gives clear experimental protocol: fixed preprocessing, same embeddings, same optimizer/loss/epochs, identical label space—reduces confounds in the comparison.

**Weaknesses:**

1.	The paper limits to CNN and BiLSTM with frozen GloVe and leaves out standard contextual models like BERT, RoBERTa, or lightweight transformer variants. It is impossible to tell whether the reported numbers are competitive or simply reflect older baselines without those comparisons. Also, transformers tend to capture long-range context, subtle semantics, and common obfuscations, so the main claims aren’t well anchored to the current literature.
2.	Splits are described only as 90/5/5 with no indication of user-disjoint or time-based partitioning, increasing the risk that near-duplicate or user-correlated tweets appear across splits and inflate performance
3.	All evidence appears to come from a single dataset and single-run results, with no multi-seed repeats, confidence intervals, or significance testing. That means the small gaps between CNN and BiLSTM could just be noise. The critical hyperparameters are under-specified (conv filter sizes/counts, LSTM hidden size, dropout rates, batch size, early stopping, LR schedule).
4.	The central contribution of this paper, CNN vs. BiLSTM with fixed GloVe, revisits a comparison that’s been explored extensively in prior work. Framing the paper around legacy architectures, without expanding the methodological lens (e.g., calibration, domain shift, richer error analysis), makes the work feel more educational than forward-looking, with limited new insight for current research.

**Questions:**

1.	Why keep GloVe frozen? Did the authors test fine-tuning the embeddings or domain-specific GloVe trained on social media? Could the authors give a small ablation (frozen vs. trainable) to clarify whether gains stem from architecture or representation.
2.	Could the authors report filter widths/filters, hidden sizes, dropout, batch size, learning-rate schedule, early stopping, and hardware/time. Also, did you run multiple seeds? If so, could the authors share mean±std or any significance tests.
3.	The paper calls the dataset balanced (Figure 1) yet cites class-imbalance effects in limitations. Can the authors quantify the imbalance and explain the observed confusions in Non-bullying/Gender/Ethnicity (seen in Figures 2–3)?

---

### Official Review · Reviewer_v4QH · 2025-11-01

**Soundness:** 1
**Presentation:** 2
**Contribution:** 1
**Rating:** 2
**Confidence:** 5

**Summary:**

The paper presents a basic comparison of two deep learning models, GloVe+CNN and GloVe+LSTM, for multiclass cyberbullying detection across five categories: religion, age, gender, ethnicity, and non-bullying using a publicly available dataset "Cyberbullying Classification dataset”. Using pre-trained GloVe embeddings for feature representation, the study found that GloVe+LSTM outperformed GloVe+CNN in accuracy, F1-score, and recall, especially for minority classes like ethnicity and gender.  While LSTM demonstrated better contextual understanding, CNN trained faster and performed well on dominant classes, making it suitable for real-time applications.
The literature is full of this kind of studies and paper severely lacks in contributions.
Lot of typos, roughly written paper.

**Strengths:**

Shows importance of sequential models

**Weaknesses:**

no contribution, reiterated the findings of other with a different dataset.

**Questions:**

What are your contributions which solves the problem in an improved way?

---

### Meta-Review · Area_Chair_UQct · 2026-01-07

**Summary:**

The paper presents a comparative study of two deep learning architectures (GloVe+CNN and GloVe+LSTM) for multiclass cyberbullying detection across five categories: religion, age, gender, ethnicity, and non-bullying, using the publicly available Cyberbullying Classification dataset. Leveraging pre-trained GloVe embeddings for feature representation, the results indicate that the GloVe+LSTM model consistently outperforms the GloVe+CNN model in terms of accuracy, F1-score, and recall, particularly for underrepresented classes such as ethnicity and gender. This performance advantage suggests that LSTMs are better at capturing long-range contextual dependencies. In contrast, the CNN-based model demonstrates faster training times and competitive performance on majority classes, making it more suitable for time-sensitive or real-time applications.

Despite these findings, the paper requires substantial polishing in terms of presentation, clarity, and experimental rigor. Moreover, the study does not meet ICLR standards with respect to novelty and technical depth, as it primarily evaluates well-established architectures on an existing dataset without introducing new modeling insights, methodologies, or analyses.

**Reviewer Concerns:**

Authors didn't address the reviews.

**Reviewer Scores:**

I don't think any reviewers will change their mind.

---

### Decision · Program_Chairs · 2026-01-26

Reject